# RoboEXP: Action-Conditioned Scene Graph via Interactive Exploration for Robotic Manipulation

Hanxiao Jiang[1]    Binghao Huang[1]    Ruihai Wu[3]    Zhuoran Li[4]
Shubham Garg[2]    Hooshang Nayyeri[2]    Shenlong Wang[1]    Yunzhu Li[1]

[1]University of Illinois Urbana-Champaign    [2]Amazon    [3]Peking University    [4]National University of Singapore

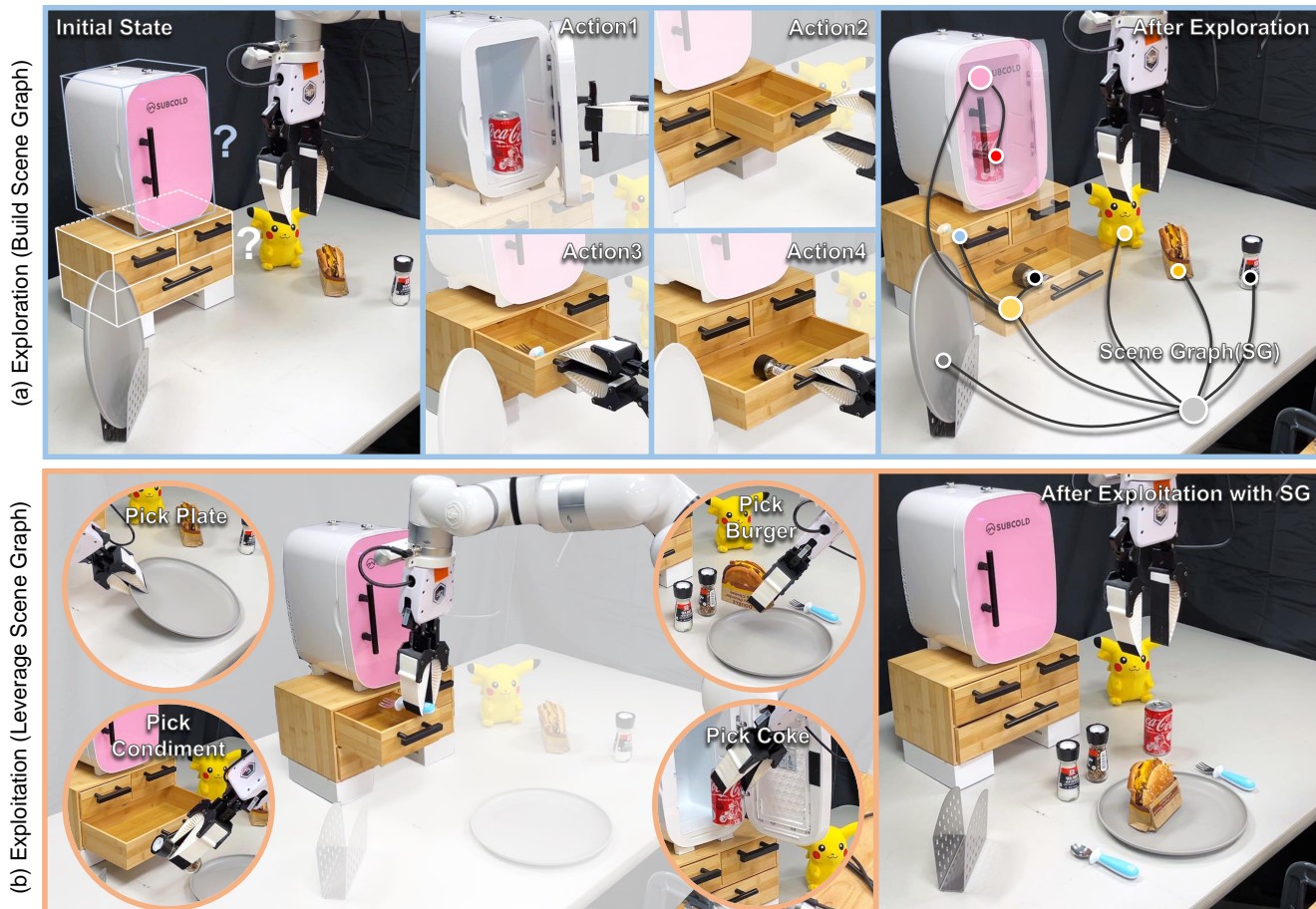

Figure 1. **Interactive Exploration to Construct an Action-Conditioned Scene Graph (ACSG) for Robotic Manipulation.** (a) **Exploration:** The robot autonomously explores by interacting with the environment to generate a comprehensive ACSG. This graph is used to catalog the locations and relationships of items. (b) **Exploitation:** Utilizing the constructed scene graph, the robot completes downstream tasks by efficiently organizing the necessary items according to the desired spatial and relational constraints.

## Abstract

*Robots need to explore their surroundings to adapt to and tackle tasks in unknown environments. Prior work has proposed building scene graphs of the environment but typically assumes that the environment is static, omitting regions that require active interactions. This severely limits their ability to handle more complex tasks in household and office environments: before setting up a table, robots must explore drawers and cabinets to locate all utensils and condiments. In this work, we introduce the novel task of interactive scene exploration, wherein robots autonomously explore environments and produce an action-conditioned scene graph (ACSG) that captures the structure of the underlying environment. The ACSG accounts for both low-level information, such as geometry and semantics, and high-level information, such as the action-conditioned relationships between different entities in the scene. To this end, we present*

*the Robotic Exploration (RoboEXP) system, which incorporates the Large Multimodal Model (LMM) and an explicit memory design to enhance our system's capabilities. The robot reasons about what and how to explore an object, accumulating new information through the interaction process and incrementally constructing the ACSG. We apply our system across various real-world settings in a zero-shot manner, demonstrating its effectiveness in exploring and modeling environments it has never seen before. Leveraging the constructed ACSG, we illustrate the effectiveness and efficiency of our RoboEXP system in facilitating a wide range of real-world manipulation tasks involving rigid, articulated objects, nested objects like Matryoshka dolls, and deformable objects like cloth.*

## 1. Introduction

Imagine a future household robot designed to prepare breakfast. This robot must efficiently perform various tasks such as conducting inventory checks in cabinets, fetching food from the fridge, gathering utensils from drawers, and spotting leftovers under food covers. Key to its success is the ability to interact with and explore the environment, especially to find items that aren't immediately visible. Equipping it with such capabilities is crucial for the robot to effectively complete its everyday tasks.

Robot exploration and active perception have long been challenging areas in robotics [1–16]. Various techniques have been proposed, including information-theoretic approaches, curiosity-driven exploration, frontier-based methods, and imitation learning [1, 13–15, 17–25]. Nevertheless, previous research has primarily focused on exploring static environments by merely changing viewpoints in a navigation setting or has been limited to interactions with a small set of object categories, such as drawers, or a closed set of simple actions like pushing [26].

In this work, we investigate the interactive scene exploration task, where the goal is to efficiently identify all objects, including those that are directly observable and those that can only be discovered through interaction between the robot and the environment (see Fig. 1). Towards this goal, we present a novel scene representation called action-conditioned 3D scene graph (ACSG). Unlike conventional 3D scene graphs that focus on encoding static relations, ACSG encodes both spatial relationships and logical associations indicative of action effects (e.g., opening a fridge will reveal an apple inside). We then show that interactive scene exploration can be formulated as a problem of action-conditioned 3D scene graph construction and traversal.

Tackling interactive scene exploration poses challenges: how can we reason about which objects need to be explored, choose the right action to interact with them, and maintain knowledge about our exploration findings? With these

challenges in mind, we propose a novel, real-world robotic exploration framework, the RoboEXP system. RoboEXP can handle diverse exploration tasks in a zero-shot manner, constructing complex action-conditioned 3D scene graph in various scenarios, including those involving obstructing objects and requiring multi-step reasoning (Fig. 2). We evaluate our system across various settings, spanning simple, single-object scenarios to complex environments, demonstrating its adaptability and robustness. The system also effectively manages different human interventions. Moreover, we show that our reconstructed action-conditioned 3D scene graph demonstrates strong capacity in performing multiple complex downstream tasks. Action-conditioned 3D scene graph advances LLM/LMM-guided robotic manipulation and decision-making research [27, 28], extending their operation domain from environments with known or observable objects to complicated environments with unknown or unobserved ones. To our knowledge, this is the first of its kind.

Our contributions are as follows: i) we propose action-conditioned 3D scene graph and introduce the interactive scene exploration task to address the challenging interaction aspect of exploration; ii) we develop the RoboEXP system, capable of exploring complicated environments with unseen objects in a wide range of settings; iii) through extensive experiments, we demonstrate our system's ability to construct complex and complete action-conditioned 3D scene graph, demonstrating significant potential for various manipulation tasks. Our experiments involve rigid and articulated objects, nested objects like Matryoshka dolls, and deformable objects like cloth, showcasing the system's generalization ability across objects, scene configurations, and downstream tasks.

## 2. Related Works

**Scene graphs** [29, 30] represent objects and their relations [31–33] in a scene via a graph structure. Previous studies generate scene graphs from images [30, 34] or 3D scenes [35] with hierarchical and semantic information, and further with the assistance of large language models (LLMs) [36]. They leverage scene graphs for image captioning [37, 38], image retrieval and generation [29, 39], visual-language tasks [31, 40], navigation [41, 42] and task planning [43–45]. While previous works model scene graphs in static 2D or 3D scenes, we generate action-conditioned scene graphs that integrate actions as core elements, depicting interactive relationships between objects and actions. This action-centric approach opens avenues for physical exploration and diverse downstream robotics tasks.

**Robotic exploration** aims to autonomously navigate, interact with, and gather information from environments it has never encountered before. It is applicable in search and rescue [1, 2, 46–52], planetary exploration [3, 4, 53, 54], ob-

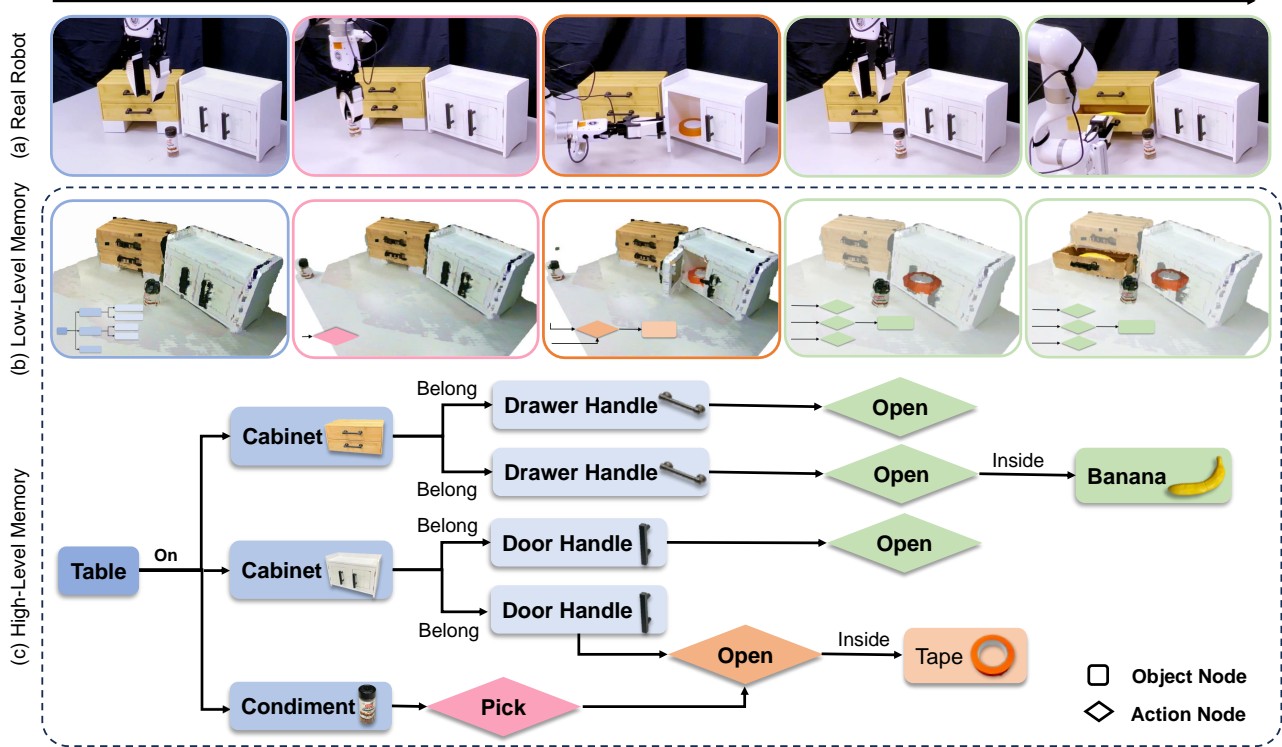

Figure 2. **Action-Conditioned 3D Scene Graph from Interactive Scene Exploration.** To illustrate the construction process of our ACSG in the interactive scene exploration, we depict a scenario wherein a robot arm explores a tabletop scene containing two cabinets and a condiment obstructing the left door. (a) The robot arm actively interacts with the scene, completing the interactive scene exploration process. (b) We showcase the corresponding low-level memory in our ACSG, which represents the geometry and semantic information of the scene. The small graph within each visualization represents a segment of the final scene graph. (c) We present the high-level memory of our action-conditioned scene graph. The graph reveals that picking up the condiment serves as a precondition for opening the door, and opening the bottom drawer allows the observation of the concealed tape and banana.

ject goal navigation [5, 6, 55–72], and mobile manipulation [7, 8, 73–76]. The primary guiding principle behind robotic exploration is to reduce the uncertainty of the environment [17–19, 46, 77, 78], making uncertainty quantification key for robotic exploration tasks. Curiosity-driven exploration has recently emerged as a promising approach, showing effective results in various contexts [15, 20, 21, 79]. Most past works have focused on exploration in the context of mobility [1, 2, 5–8, 46–50, 55–69, 73–76, 80], with the primary goal of modeling and understanding the static environment to complete specific tasks. Recently, exploration has also been studied in the context of manipulation [16, 23, 81–84], aiming to better understand the scene by changing the state of the environment. Our work introduces a new active exploration strategy for manipulation, uniquely defining a novel scene graph-guided objective to guide the exploration process.

## 3. Problem Statement

We unfold this section with an introduction of action-conditioned 3D scene graph, a novel scene representation illustrating interactive object relationships (Sec. 3.1). We

then formulate interactive scene exploration as an action-conditioned 3D scene graph construction and traversal problem (Sec. 3.2).

### 3.1. Action-Conditioned 3D Scene Graph

An action-conditioned 3D scene graph (ACSG) is an actionable, spatial-topological representation that models objects and their interactive and spatial relations in a scene. Formally, ACSG is a directed acyclic graph $\mathbf{G} = (\mathbf{V}, \mathbf{E})$ where each node represents either an object (e.g., a *door*) or an action (e.g., *open*), and edges $\mathbf{E}$ represent their interaction relations. The object node $\mathbf{o}_i = (\mathbf{s}_i, \mathbf{p}_i) \in \mathbf{V}$ encodes the semantics and geometry of each object (e.g., the semantic embedding of a fridge $\mathbf{s}_i$, and its shape in the form of a point cloud $\mathbf{p}_i$), whereas the action node $\mathbf{a}_k = (a_k, \mathbf{T}_k) \in \mathbf{V}$ encodes high-level action type $a_k$ and low-level primitives $\mathbf{T}_k$ to perform the actions. Between the nodes are edges encoding their relations, which we categorize into four types: 1) between objects $\mathbf{e}_{\mathbf{o} \to \mathbf{o}}$ (e.g., the *door handle belongs* to the *fridge*), 2) from objects to actions $\mathbf{e}_{\mathbf{o} \to \mathbf{a}}$ (e.g., *toy* can be *picked up*), 3) from action to objects $\mathbf{e}_{\mathbf{a} \to \mathbf{o}}$ (e.g., a *banana*

can be reached if we *open* the cabinet), or 4) from one action to another $\mathbf{e_{a \to a}}$ (e.g., the cabinet can be *opened* only if we *move away* the *condiment*). Our action-conditioned 3D scene graph greatly enhances existing 3D scene graphs, as it explicitly models the action-conditioned relations between objects. Fig. 2 depicts a complete action-conditioned 3D scene graph of a tabletop scene.

One advantage of our interaction-aware scene graph lies in its simplicity for retrieving and taking actions on an object. Regardless of how complicated the scene is, given our scene graph and a target object, an agent merely needs to sequentially execute all the actions on the paths from the root to the object node in a topological order to retrieve the object. For example, in Fig. 2, to reach the tape inside a cabinet whose door is blocked by a condiment, according to the graph, one simply needs to: 1) pick up the condiment on the table that blocks the cabinet door, and 2) open the cabinet through the door handle.

### 3.2. Interactive Exploration

This subsection describes how we can construct a complete action-conditioned scene graph of a real-world scene. This is a challenging problem due to partial observability. For instance, a banana cannot be populated without *opening* the cabinet. To solve this task, we formulate the scene graph construction as an active perception and exploration problem using POMDP-inspired notations. Formally, at each time $t$, based on our past graph estimation $\mathbf{G}^{t-1}$, and past sensor observations $\mathbf{O}^{t-1}$, our agent takes an action $\mathbf{A}^t$, which causes the environment to transition to a new state, and the agent receives a new observation $\mathbf{O}^t$, which is used to update its current inferred graph $\mathbf{G}^t$. This update might include adding new nodes to the graph or updating the state of an existing node. We will then continue with exploration and keep updating the set of remaining unexplored nodes $\mathbf{U} \subset \mathbf{V}$ (see Algorithm 1).

The goal of the exploration is simple: discover and explore all the nodes of the scene graph in as little time as possible. Towards this, we formulate a reward function with three terms:

$$\mathbf{R}^t = \mathbf{R}^t_{\text{graph}} + \mathbf{R}^t_{\text{explore}} + \mathbf{R}^t_{\text{time}}$$

where $\mathbf{R}^t_{\text{graph}} = |\mathbf{V}^t| - |\mathbf{V}^{t-1}|$ is the graph construction term, which promotes our agent to discover as many nodes as possible to the graph, $\mathbf{R}^t_{\text{explore}} = \max(0, |\mathbf{U}^{t-1}| - |\mathbf{U}^t|)$ gives positive reward to actions that reduce unexplored node set, which prioritize the agent to explore previously unexplored nodes, and immediate reward $\mathbf{R}^t_{\text{time}} = -\lambda, 0 < \lambda < 1$ is a negative time reward that optimizes the time efficiency and allows the exploration to terminate when there is no more node to explore.

Intuitively, to maximize this reward at each discrete timestamp, we should prioritize exploring the unexplored nodes

---

**Algorithm 1** Interactive Exploration

```
 1: input: O⁰, G⁰ = (V⁰, E⁰), U⁰ ← V⁰
 2: while |Uᵗ⁻¹| ≠ 0 do
 3:    if choose object oᵢ ∈ Uᵗ⁻¹ then        % explore object
 4:       add spatial relations                      % memory
 5:       obtain action a to explore oᵢ      % decision-making
 6:       if action a ∉ Vᵗ⁻¹ then
 7:          Vᵗ, Uᵗ = Vᵗ⁻¹ ∪ {a}, Uᵗ⁻¹ ∪ {a}   % add node
 8:          Eᵗ = Eᵗ⁻¹ ∪ {e_{oᵢ→a}}                % add edge
 9:          Uᵗ = Uᵗ \ oᵢ                % mark as explored
10:       end if
11:    else choose action a_k ∈ Uᵗ⁻¹
12:       if no obstruction then             % decision-making
13:          take action a_k                          % action
14:          obtain new observation Oᵗ            % perception
15:          if found new objects O ⊄ Vᵗ⁻¹ then
16:             Vᵗ, Uᵗ = Vᵗ ∪ {O}, Uᵗ⁻¹ ∪ {O}  % add nodes
17:             Eᵗ = Eᵗ ∪ {e_{a_k→O}}               % add edges
18:             Uᵗ = Uᵗ \ a_k              % mark as explored
19:          end if
20:       else
21:          add action preconditions                % memory
22:       end if
23:    end if
24: end while
25: output: Gᵗ                        % final scene graph
```

---

in the current scene graph that are likely to lead to the discovery of new nodes (e.g., opening a cabinet that has not been opened, or lifting a piece of clothing that might cover a small object). The key challenge lies in how we can perceive the objects in the scene, infer possible actions and their relations from the sensory data, and take actions with the current scene graph. In the next section, we will comprehensively describe our system implementation to achieve this goal.

## 4. Method

To tackle the task outlined in Section Sec. 3, we present our RoboEXP system, designed to autonomously explore unknown environments by observing and interacting with them. The system comprises four key components: perception, memory, decision-making, and action modules (see Fig. 3). Raw RGBD images are captured through the wrist camera in different viewpoints and processed by the perception modules to extract scene semantics, including object labels, 2D bounding boxes, segmentations, and semantic features. The obtained semantic information is then transmitted to the memory module, where the 2D data is merged into the 3D representation. Such 3D information serves as a valuable guide for the decision module, aiding in the selection of appropriate actions to further interact or observe the environment and unveil hidden objects. The action module is activated to execute the planned action, generating new

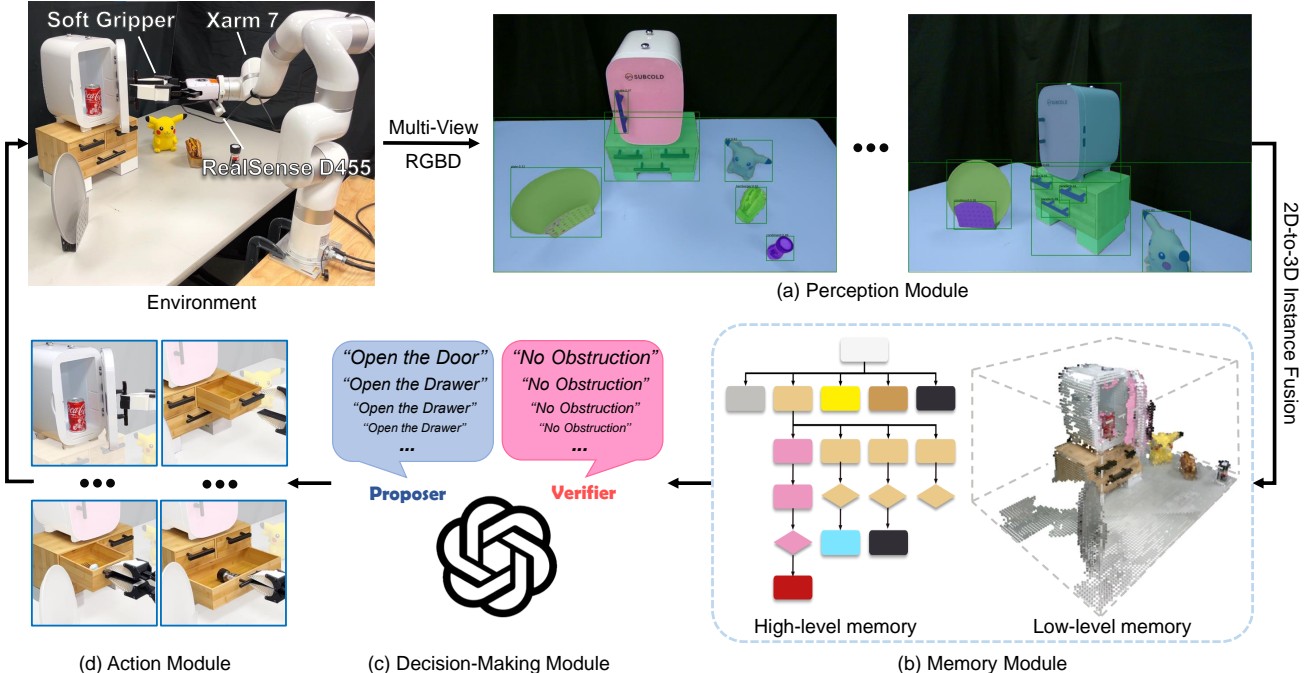

Figure 3. **Overview of Our RoboEXP System.** We present a comprehensive overview of our RoboEXP system, comprised of four modules. (a) Our **perception module** takes RGBD images as input and produces the corresponding 2D bounding boxes, masks, object labels, and associated semantic features as output. (b) The **memory module** seamlessly integrates 2D information into the 3D space, achieving more consistent 3D instance segmentation. Additionally, it constructs the high-level graph of our ACSG through the merging of instances. (c) Our **decision-making module** serves dual roles as a proposer and verifier. The proposer suggests various actions, such as opening doors and drawers, while the verifier assesses the feasibility of each action, considering factors like obstruction. (d) The **action module** executes the proposed actions, enabling the robot arm to interact effectively with the environment.

observations for the perception modules. This closed-loop system ensures the thoroughness of our task in interactive scene exploration.

**Perception Module.** Given multiple RGBD observations from different viewpoints, the objective of the perception module (Fig. 3a) is to detect and segment objects while extracting their semantic features. To enhance generality, we opt for the open-vocabulary detector GroundingDINO [85] and the Segment Anything in High Quality (SAM-HQ) [86], an advanced version of SAM [87]. For the extraction of semantic features used in subsequent instance merging within the memory module, we employ CLIP [88]. To obtain per-instance CLIP features, we implement a strategy similar to the one proposed by Jatavallabhula et al. [89]. Specifically, we extend the local-global image feature merging approach by incorporating additional label text features to augment the semantic CLIP feature for each instance. Furthermore, we exclusively focus on instance-level features, disregarding pixel-level features, thereby accelerating the entire semantic feature extraction process.

**Memory Module.** The memory module (Fig. 3b) is designed to construct our ACSG of the environment by assimilating observations over time. For the low-level memory, to ensure stable instance merging from 2D to 3D, we employ a similar instance merging strategy as presented in Lu et al. [90], consolidating observations from diverse RGBD sources across various viewpoints and time steps. In contrast to the original algorithm, which considers only 3D IoU and

semantic feature similarity we additionally incorporate label similarity and instance confidence. To enhance algorithm efficiency, we represent low-level memory using a voxel-based representation, which allows for more efficient computation and memory updates. Meanwhile, given the crowded nature of objects in our tabletop setting, we have implemented voxel-based filtering designs to obtain a cleaner and more complete representation of the objects for storage in our memory.

The memory module handles merging across different viewpoints and time steps. To merge across different viewpoints, we project 2D information (RGBD, semantic features, mask, bounding box) to 3D and leverage the instance merging strategy mentioned earlier to attain consistent 3D information. Addressing memory updates across time steps presents a challenge due to dynamic changes in the environment. For instance, a closed door in the previous time step may be opened by our robot in the current time step. To accurately reflect such changes, our algorithm evaluates whether elements within our memory have become outdated, primarily through depth tests based on the most recent observations. This process ensures that the memory accurately represents the environment's current state, effectively managing scenarios where objects may change positions or states across different time steps.

For the high-level graph of our ACSG, the memory module analyzes the relationships between objects and the logical associations between actions and objects. Depending on

Table 1. **Quantitative Results on Different Tasks.** We compare the performance of both the GPT-4V baseline and our system across various tasks. We assess the outcomes using five distinct metrics to illustrate diverse facets of the interactive exploration process. Our system consistently outperforms the baseline across all tasks and metrics.

| Task (10 variance for each) | Drawer-Only | | Door-Only | | Drawer-Door | | Recursive | | Occlusion | |
|---|---|---|---|---|---|---|---|---|---|---|
| Metric | GPT-4V | Ours | GPT-4V | Ours | GPT-4V | Ours | GPT-4V | Ours | GPT-4V | Ours |
| Success % ↑ | 20±13.3 | **90**±10.0 | 30±15.2 | **90**±10.0 | 10±10.0 | **70**±15.3 | 0±0.0 | **70**±15.3 | 0±0.0 | **50**±16.7 |
| Object Recovery % ↑ | 83±11.0 | **97**±3.3 | 50±16.7 | **100**±0.0 | 62±10.7 | **91**±4.7 | 20±13.3 | **80**±11.7 | 17±11.4 | **67**±14.9 |
| State Recovery % ↑ | 60±16.3 | **100**±0.0 | 80±13.3 | **100**±0.0 | 70±15.3 | **100**±0.0 | 70±15.3 | **100**±0.0 | 10±10.0 | **70**±15.3 |
| Unexplored Space % ↓ | 15±7.6 | **0**±0.0 | 40±14.5 | **0**±0.0 | 25±6.5 | **0**±0.0 | 63±15.3 | **15**±8.9 | 85±7.6 | **30**±15.3 |
| Graph Edit Dist. ↓ | 2.8±1.04 | **0.2**±0.20 | 4.4±1.42 | **0.1**±0.10 | 5.6±1.46 | **0.5**±0.27 | 8.8±2.06 | **2.1**±1.49 | 7.3±0.97 | **2.5**±1.15 |

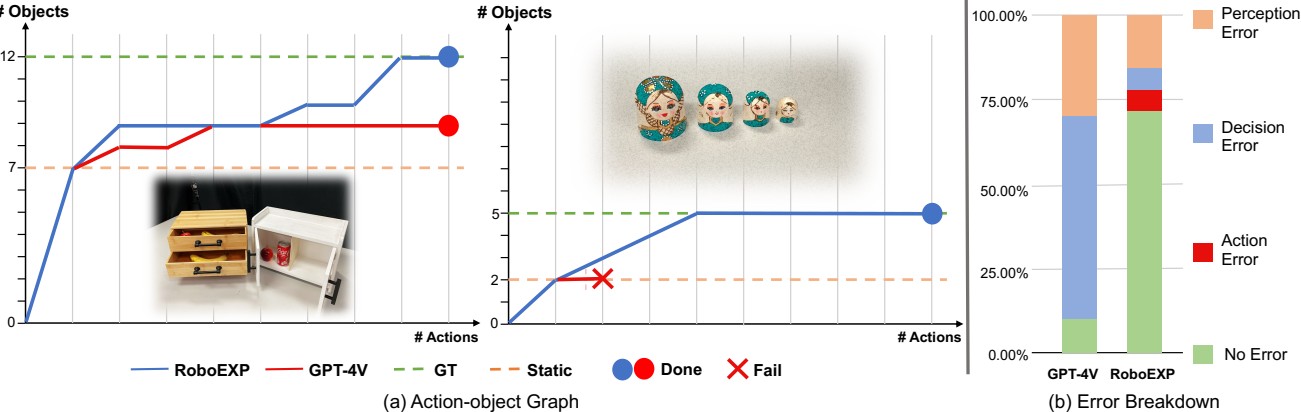

(a) Action-object Graph               (b) Error Breakdown

Figure 4. **Visualization of Quantitative Results.** (a) The action-object graph captures the change in the number of discovered objects relative to the number of actions taken. Our RoboEXP efficiently discovers all objects. Sometimes, the object count doesn't increase during actions due to the absence of objects in storage after opening. Additionally, some actions are employed to restore the scene state (e.g., closing the door after exploration). (b) The error breakdown of all our quantitative experiments includes 5 task settings with 10 variations each. We categorize errors into perception, decision, action, and no-error cases. For the GPT-4V baseline, manual assistance in action execution eliminates failure cases, serving as an upper bound for baseline performance. Even in this scenario, our RoboEXP largely outperforms the baseline.

changes in low-level memory and relationships, the memory module is tasked with updating the graph. This involves adding, deleting, or modifying nodes and edges within our graph.

**Decision-Making Module.** The primary goal of the decision module (Fig. 3c) is to identify the appropriate object and corresponding skill to enhance the effectiveness and efficiency of interactive scene exploration. In the context of our task, distinct objects may necessitate distinct exploration strategies. While humans can easily discern the most suitable skill to apply (e.g., picking up the top Matryoshka doll to inspect its contents), achieving such decisions through heuristic-based methods is challenging. The utilization of a Large Multi-Modal Model (LMM), such as GPT-4V [91, 92], shows instrumental in addressing this difficulty, as it captures commonsense knowledge that facilitates decision-making.

The LMM brings commonsense knowledge to our decision-making process and serves in two pivotal roles. Firstly, it functions as an action proposer. Given the current digital environment from the memory module, GPT-4V is tasked with selecting the appropriate skill for unexplored objects in our system. For instance, when presented with a visual prompt of an object within a green bounding box from various viewpoints, GPT-4V can discern the suitable "pick up" skill for the Matryoshka doll in the environment. For unexplored objects, our ACSG includes the attribute of whether each object node is explored or unexplored. GPT-4V, in its role as the proposer, also functions to assess whether the object holds value for further exploration. If not, the corresponding node is marked as explored, indicating that no further actions are needed.

Secondly, the LMM also serves as the action verifier. For the proposer role, it analyzes the object-centric attributes and doesn't consider surrounding information when choosing the proper skill. For example, if the proposed action involves opening a door, the proposer alone may struggle with cases where obstructions exist in front of the door (e.g., a condiment bottle). To address this, we use another LMM program to verify the feasibility of the action and identify any objects in the scene that may impede the action based on information from our ACSG.

In summary, the decision module, with its dual roles, effectively guides our system to choose efficient actions that minimize uncertainty in the environment and successfully

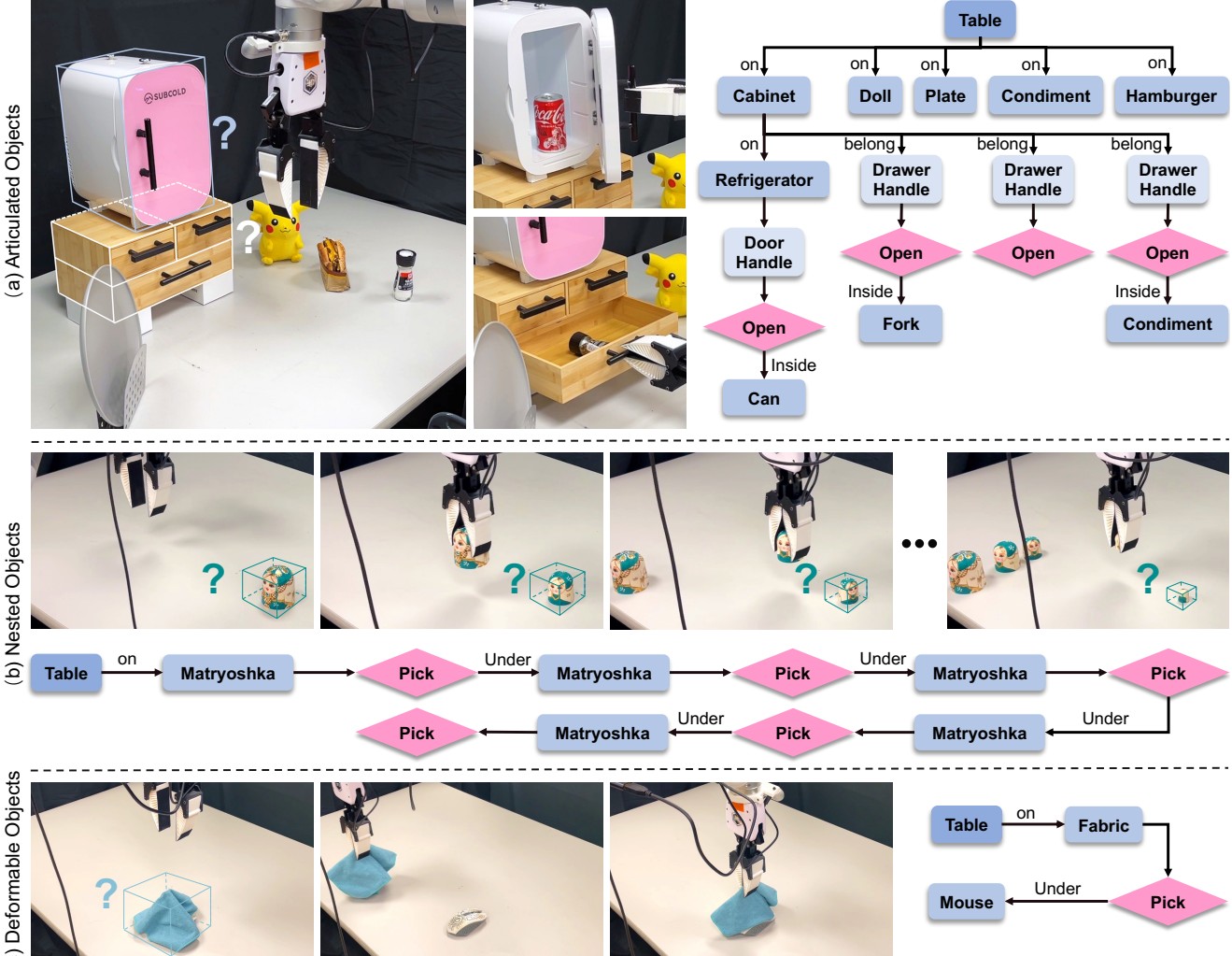

Figure 5. **Qualitative Results on Different Scenarios.** We visualize the interactive exploration process and the corresponding constructed ACSG. (a) This scenario involves a tabletop environment with two articulated objects, accompanied by additional items either on the table or concealed in storage space. The constructed scene graph demonstrates the success of our system in identifying all objects within the environment through a series of physical interactions. (b) This scenario includes nested objects, five Matryoshka dolls, with only the top one being directly observable. Our system autonomously decides to explore the contents through a recursive reasoning process, showcasing its ability to construct deep ACSG. (c) This scenario involves a fabric covering a mouse, showcasing exploration scenarios that involve a deformable object. Our system interacts with the fabric and successfully uncovers what lies beneath it.

locate all relevant objects.

**Action Module.** In the action module (Fig. 3d), our primary focus is on autonomously constructing the ACSG through effective and efficient interaction with the environment. We employ heuristic-based action primitives within our action module, leveraging the geometry cues in our ACSG. These primitives encompass seven categories: "open the door", "open the drawer", "close the door", "close the drawer", "pick object to idle space", "pick back object", "move wrist camera to position". Strategic utilization of these skills plays a pivotal role in accomplishing intricate tasks seamlessly within our system (more details in the Appendix).

# 5. Experiments

In this section, we assess the performance of our system across a variety of tabletop scenarios in the interactive scene exploration setting. Our primary objective is to address two key questions through experiments: 1) How does our system effectively and efficiently deal with diverse exploration scenarios and successfully construct comprehensive ACSG? 2) What is the utility of our ACSG in facilitating downstream tasks?

## 5.1. Interactive Exploration and Scene Graph Building

To assess our system's efficacy across various exploration scenarios, we compared it with a strong baseline by augment-

ing GPT-4V with ground truth actions. We designed five types of experiments, each with 10 different settings varying in object number, type, and layout. Our quantitative analysis reveals that our RoboEXP system consistently surpasses the baseline across various tasks. Furthermore, we validate the performance of our system in constructing ACSG through qualitative demonstrations. Check the Appendix and our supplementary video for more details.

**Evaluation.** To thoroughly assess the efficacy of our system compared to the baseline, we have designed five key metrics (Success, Object Recovery, State Recovery, Unexplored Space, Graph Edit Distance) to measure its performance. It is crucial to note that the output of our task, represented by ACSG, aligns precisely with the format of ACSG for our system. Conversely, for the baseline, we manually construct ACSG based on its actions and the new observations it uncovers. Due to the unstructured nature of the raw scene graph from the baseline, we carefully refine it according to the observable objects, providing an upper-bound baseline for comparison during evaluation.

**Comparison.** The quantitative findings presented in Table 1 underscore the superior performance of our system compared to the baseline method. Our approach showcases a notable enhancement across all metrics, outperforming the baseline by a considerable margin. The collective assessment of success rate, object recovery, and unexplored space metrics unequivocally validates the efficacy of our system in exploring unfamiliar scenes through interactive processes. It is essential to highlight that in the case of object recovery, the baseline method may occasionally choose to randomly open certain drawers or doors to unveil objects. This randomness contributes to a seemingly higher object recovery rate for the baseline, which may not necessarily correlate with its overall success. The unexplored space metric shows that our system is much more stable in exploring all need-to-explore spaces.

Moreover, both the success rate and graph edit distance underscore the close alignment of our system with human actions, highlighting the efficiency of our approach across diverse scenarios. The state recovery metric assesses whether the final state post-exploration resembles the initial state. Our system consistently shows effective state recovery; however, the baseline may trick this metric by opting not to take any action, resulting in an artificially high score in this aspect.

Fig. 4a provides additional insights, illustrating that as the number of actions increases, so does the number of objects. Specifically, we present the ground truth object number alongside the directly-observable object number that can be represented by the traditional 3D scene graph. These results underscore our system's ability to achieve robust and efficient exploration throughout the exploration process. Our system excels in efficiently discovering all concealed objects,

whereas the baseline fails either due to a lack of early-stage actions or an inability to explore all need-to-explore spaces even upon completion. The analysis of errors (Fig. 4b) in both our system and the baseline reveals the specific failure cases encountered by the baselines. In contrast, our system demonstrates enhanced robustness in both perception and decision-making.

Fig. 5 further illustrates various exploration scenarios along with their corresponding ACSG. These scenarios encompass ACSG with varying width or depth, highlighting our system's adaptive capability across diverse objects such as rigid, articulated objects, nested objects, and deformable objects. In addition, the scenario in Fig. 2 shows that our system is able to deal with the scenario with obstruction.

## 5.2. Utility of our ACSG

The scenarios depicted in Fig. 1 exemplify the efficacy of our generated output (ACSG) in manipulation tasks. Consider the table-rearranging scenario: without our ACSG, the robot struggles to swiftly prepare the table due to the lack of precise prior knowledge about the location of objects (e.g., the fork stored in the top-left drawer of the wooden cabinet). Beyond comprehensive layout guidance, our ACSG also addresses a crucial question regarding task feasibility for the robot. For instance, if there is no spoon in the scene, the robot recognizes its inability to perform the task and asks for human help.

In addition to enhancing downstream manipulation tasks, our ACSG possesses the capability to autonomously adapt to environmental changes. In the human intervention setting, our system seamlessly explores newly added components, such as a cabinet, ensuring continuous adaptability. Check our Appendix and supplemental video for more details.

## 6. Conclusion

We introduced RoboEXP, a foundation-model-driven robotic exploration framework capable of effectively identifying all objects in a complex scene, both directly observable and those revealed through interaction. Central to our system is action-conditioned 3D scene graph, an advanced 3D scene graph that goes beyond traditional models by explicitly modeling interactive relations between objects. Experiments have shown RoboEXP's superior performance in interactive scene exploration across various challenging scenarios, significantly outperforming a strong GPT4V-based baseline. Notably, the reconstructed action-conditioned 3D scene graph is crucial for guiding complex downstream manipulation tasks, like preparing breakfast in a mock-kitchen environment with fridges, cabinets, and drawer sets. Our system and its action-conditioned scene graph lay the groundwork for practical robotic deployment in complex settings, especially in environments like households and offices, facilitating their everyday use.

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
