# OpenReview forum: "RoboEXP: Action-Conditioned Scene Graph via Interactive Exploration for Robotic Manipulation"
_thecvf.com/CVPR/2024/Workshop/VLADR — VLADR 2024 Oral_

### Official Review · Reviewer_MGu2 · 2024-04-20

**Rating:** 9
**Confidence:** 3

**Review:**

The Robotic Exploration (RoboEXP) system, powered by the Large Multimodal Model (LMM) and an advanced memory framework, navigates diverse real-world environments with unprecedented adaptability. Without prior training, it intelligently explores and maps unfamiliar territories, gradually constructing an Abstract Conceptual Spatial Graph (ACSG). This innovative approach enables RoboEXP to efficiently tackle a spectrum of manipulation tasks, from handling rigid and articulated objects to navigating nested structures like Matryoshka dolls and manipulating deformable materials such as cloth. Given all, I would like to give `accept`.

---

### Decision · Program_Chairs · 2024-04-22

Accept (Oral)